# Diagnostic Accuracy of Deep Learning for the Prediction of Osteoporosis Using Plain X-rays: A Systematic Review and Meta-Analysis

**DOI:** 10.3390/diagnostics14020207

**Published:** 2024-01-18

**Authors:** Tzu-Yun Yen, Chan-Shien Ho, Yueh-Peng Chen, Yu-Cheng Pei

**Affiliations:** 1Department of Physical Medicine and Rehabilitation, Chang Gung Memorial Hospital, Linkou No. 5, Fuxing Street, Guishan District, Taoyuan City 333, Taiwan; yenhsnu42@cgmh.org.tw (T.-Y.Y.); longarea@cgmh.org.tw (C.-S.H.); 2School of Medicine, Chang Gung University, No. 259, Wenhua 1st Road, Guishan District, Taoyuan City 333, Taiwan; 3Center for Artificial Intelligence in Medicine, Chang Gung Memorial Hospital, Linkou No. 5, Fuxing Street, Guishan District, Taoyuan City 333, Taiwan; yuepengc@cgmh.org.tw; 4Master of Science Degree Program in Innovation for Smart Medicine, Chang Gung University, No. 259, Wenhua 1st Road, Guishan District, Taoyuan City 333, Taiwan; 5Center of Vascularized Tissue Allograft, Gung Memorial Hospital, Linkou No. 5, Fuxing Street, Guishan District, Taoyuan City 333, Taiwan

**Keywords:** osteoporosis, osteopenia, bone mineral density, convolutional neural network, deep learning, X-ray

## Abstract

(1) Background: This meta-analysis assessed the diagnostic accuracy of deep learning model-based osteoporosis prediction using plain X-ray images. (2) Methods: We searched PubMed, Web of Science, SCOPUS, and Google Scholar from no set beginning date to 28 February 2023, for eligible studies that applied deep learning methods for diagnosing osteoporosis using X-ray images. The quality of studies was assessed using the Quality Assessment of Diagnostic Accuracy Studies-2 criteria. The area under the receiver operating characteristic curve (AUROC) was used to quantify the predictive performance. Subgroup, meta-regression, and sensitivity analyses were performed to identify the potential sources of study heterogeneity. (3) Results: Six studies were included; the pooled AUROC, sensitivity, and specificity were 0.88 (95% confidence interval [CI] 0.85–0.91), 0.81 (95% CI 0.78–0.84), and 0.87 (95% CI 0.81–0.92), respectively, indicating good performance. Moderate heterogeneity was observed. Mega-regression and subgroup analyses were not performed due to the limited number of studies included. (4) Conclusion: Deep learning methods effectively extract bone density information from plain radiographs, highlighting their potential for opportunistic screening. Nevertheless, additional prospective multicenter studies involving diverse patient populations are required to confirm the applicability of this novel technique.

## 1. Introduction

Osteoporosis is a common clinical problem in older adults and a major public health issue worldwide [1]. Bone strength and structural integrity decline with age, leading to an increased risk of fragility fractures in older adults. Fragility fractures increase morbidity and mortality in individuals and impose a huge financial burden on society as a whole [2]. Considering that osteoporosis is often undetected until a fracture occurs, early detection is essential for its treatment.

According to the World Health Organization (WHO), osteoporosis is diagnosed by measuring the bone mineral density (BMD) of the femoral neck using dual-energy X-ray absorption (DXA) [3] and is defined as BMD ≥ 2.5 standard deviations (SDs) below the average value of the young white female reference population (T-score ≤  −2.5), whereas low bone mass (or osteopenia) is defined as ≥1.0 SD but <2.5 SDs below the average value of the young white female reference population (−2.5 < T-score ≤ −1.0) [4]. Despite being the gold standard for osteoporosis diagnosis, DXA has not been adequately applied in osteoporosis screening because of its low availability, high cost [5], lack of public awareness of osteoporosis [6], lack of financial incentives to promote osteoporosis screening [7], and lack of DXA prescriptions [8].

Given the low rates of DXA screening, opportunistic screening presents an intriguing solution because it utilizes images obtained for other indications and does not require additional costs, radiation exposure, or patient time. Previous studies mainly used computed tomography (CT) and magnetic resonance imaging (MRI) to estimate BMD [9], classify the degree of bone loss [10,11], and predict the risk of osteoporotic fractures [12]. For example, Pickhardt et al. demonstrated the feasibility of using abdominal CT scans to estimate the BMD value, providing an area under the receiver operating characteristic curve (AUROC) of 0.83 and a sensitivity of 0.90 in detecting osteoporosis [9]. Kadri et al. demonstrated that MRI can be used as an opportunistic screening data resource to identify patients undergoing spinal surgery who are likely to develop osteoporosis [11].

With the advancement of technology and ease of availability of medical data, computer-aided diagnostic systems for medical images (such as MRI, CT, and X-ray) have been developed to improve clinical diagnosis by providing additional information to clinicians [13]. In particular, deep learning methods are increasingly preferred over traditional machine learning techniques, mainly due to their superior capability for the efficient and automatic extraction of clinically relevant features. Moreover, their exceptional performance in processing and learning from vast, complex, and unstructured datasets further establishes deep learning as a more effective approach [14]. For osteoporosis screening, plain radiographs stand out as the more accessible and practical choice in hospitals, primarily due to their low acquisition cost, reduced examination time, and versatility in covering most body parts, which also enhances their suitability for opportunistic screening. Although research has revealed that using artificial intelligence on CT and MRI scans for BMD estimation is comparably effective to plain radiographs, with CT showing a correlation of 0.84 and an AUC of 0.97 in diagnosing osteoporosis [15], and MRI demonstrating a correlation of 0.64 [16], these methods are less practical for screening. This is because of their high costs, approximately 10 to 15 times higher than plain X-rays in Taiwan, along with limited availability and longer both waiting and examination times. To this end, several deep learning algorithms have been developed to infer the presence of osteoporosis from plain X-rays, such as dental [17,18], chest [19,20], pelvic [21,22,23,24], and lumbar [22,25] X-rays.

Lee et al. found that a CNN-based computer-aided system analyzing panoramic X-rays was highly effective in detecting osteoporosis, outperforming oral and maxillofacial radiologists with an AUC of 0.858, sensitivity of 0.9, specificity of 0.815, and an accuracy of 0.84 [18]. Wang et al. developed a method that automatically inferred the BMD of the lumbar spine from chest radiographs and showed a good correlation (R = 0.84) between the predicted and ground-truth BMD values, with an AUROC of 0.936 for detecting osteoporosis [20]. Ho et al. reported that DeepDXA, a convolutional neural network regression model analyzing pelvic X-rays, demonstrates high accuracy in diagnosing osteoporosis, showing a strong correlation (R = 0.85) between its predictions and the actual BMD measurements [21]. Zhang et al. developed a deep learning model that effectively predicts osteoporosis and osteopenia in postmenopausal women by analyzing lumbar spine X-rays, achieving an AUC of 0.767 and a sensitivity of 73.7% [25]. These studies demonstrate that plain X-rays combined with deep learning can be used for the opportunistic screening of osteoporosis.

The sample sizes of most previous studies were limited; therefore, it is necessary to conduct a meta-analysis to determine whether X-ray images are viable for the opportunistic screening of osteoporosis. Given that different studies have inferred bone density using images obtained from different body parts, it remains unclear whether the efficacy of inference differs among body parts; hence, conducting subgroup analyses is essential to determine which body part yields the best performance. In the present meta-analysis, we aimed to (1) determine the diagnostic accuracy of plain radiography using deep learning models, and (2) assess the factors that determine the diagnostic accuracy for osteoporosis.

## 2. Materials and Methods

### 2.1. Research Design

This study was conducted according to the guidelines for the Preferred Reporting Items for Systematic Reviews and Meta-Analyses of Diagnostic Test Accuracy Studies (PRISMA-DTA) [26]. The checklist is presented in the Appendix A.

### 2.2. Search Strategy

Online databases including PubMed, Web of Science, SCOPUS, and Google Scholar were used to retrieve potentially eligible studies. The literature search extended to 28 February 2023, with no beginning dates. The search terms employed included “deep learning”, “convolutional neural network”, “CNN”, “DCNN”, “osteoporosis”, “osteopenia”, “bone mineral density”, “BMD”, and “low BMD”, along with various X-ray types such as “radiographs”, “X-rays”, “diagnostic X-rays”, “chest X-rays”, “pelvic X-rays”, “lumbar spine X-rays”, and “dental X-rays”. Boolean operators like “AND” and “OR” were employed for greater search accuracy, and the scope was broadened through the use of truncations such as “radiograph*”, “osteoporos*”, and “diagnos*”. All retrieved studies were imported into EndNote and duplicate publications were automatically identified and removed. The titles and abstracts were independently reviewed by two reviewers (Y.T.Y and H.C.S), and the most relevant articles were selected for full-text review. Disagreements regarding study inclusion were resolved through discussion guided by the PRISMA guideline, and a third investigator (Y.P.C) was consulted if a consensus could not be reached.

### 2.3. Selection Criteria

Inclusion criteria for the studies were (1) the use of deep learning methods on X-ray images for osteoporosis detection or bone density estimation, (2) the utilization of DXA as the reference standard method for diagnosing osteoporosis, and (3) studies containing information about the sample sizes of the test dataset. Exclusion criteria were (1) non-English or not peer-reviewed studies; (2) abstracts, conference articles, preprints, review articles, and meta-analyses; (3) articles using the same patients for model training and model testing; (4) primary non-diagnostic accuracy (for example, intervention); and (5) studies with fewer than 30 participants in each of the training and testing datasets.

### 2.4. Quality Assessment and Risk of Bias

Two reviewers (YTY and HCS) independently assessed the quality and bias of the included studies using the Quality Assessment of Diagnostic Accuracy Studies (QUADAS-2) tool [27]. This tool includes four domains that assess the risk of bias (patient selection, index test, reference standard, flow, and timing) and three domains that assess applicability concerns (patient selection, index test, and reference standard). Each of these domains has three categories (low-, unclear-, and high-risk bias). Disagreements between the reviewers were discussed until a consensus was reached.

### 2.5. Data Extraction

Data were extracted by two independent reviewers (YTY and HCS) following the PRISMA-DTA guidelines [28]. The following data were extracted: (1) first author; (2) publication year; (3) study design and center; (4) country of origin of data used; (5) type of medical images; (6) reference diagnosis; (7) reference standard; (8) deep learning model; (9) image number; (10) amount of data in training, validation, testing, and external datasets; (11) confusion matrix; (12) sensitivity; (13) specificity; and (14) AUROC. Differences in the data extracted between the two reviewers were discussed until a consensus was reached.

### 2.6. Outcomes

The primary outcomes for the overall diagnostic accuracy of the deep learning model using X-ray images were the estimated summary sensitivity and specificity, and the AUROC calculated from the hierarchical summary receiver operating characteristic (HSROC) curve. Variables with a *p*-value < 0.05 were considered statistically significant.

### 2.7. Data Synthesis and Statistical Analysis

Review Manager 5 (RevMan 5) [Mac version, Computer program]. Version 5.4. Copenhagen: The Cochrane Collaboration, 2020 and Stata 17.0, Stata Corporation, College Station, TX, USA were used for statistical analysis. A confusion matrix was extracted for the validation dataset from each eligible study. In cases where the study did not provide an accurate number of true positive/true negative/false positive/false negative results, RevMan was used to calculate the estimated confusion matrix based on the sensitivity, specificity, and number of validation dataset images.

A coupled forest plot of sensitivity and specificity was constructed based on a bivariate binomial random effects model using RevMan [29]. We estimated the diagnostic accuracy parameters based on an HSROC model [30] using the midas command in STATA, which considers the correlations between sensitivity and specificity, as well as variability in effects across studies [31]. Furthermore, the model considers two levels of statistical distribution: within- and between-study variability. Through this approach, the model can represent variations in diagnostic accuracy and cutoff values, and identify sources of heterogeneity among diagnostic accuracy tests [32]. The HSROC curve was generated using Stata based on the bivariate model proposed by Reitsma et al. [33] because the summary ROC curve of RevMan is based on the Moses−Littenberg method, which does not provide estimates of heterogeneity between studies [34]. In the HSROC curve, sensitivity was plotted against specificity to illustrate how the sensitivity and specificity of a test were affected by different threshold levels.

In diagnostic test accuracy studies, traditional methods like Cochran’s Q test and Higgins’ I2 statistics are not suitable for assessing heterogeneity, as they do not consider the threshold effects arising from different cutoff values across studies [35,36]. Heterogeneity was identified by visually observing the asymmetry of the SROC curve and the pronounced scattering of data points from individual studies along this curve [35,37]. Subgroup, sensitivity, and meta-regression analyses were also conducted to explore the potential sources of heterogeneity, when necessary.

## 3. Results

### 3.1. Literature Search

A systematic literature search identified 190 articles from the four databases. A total of 61 articles were excluded because they were duplicates. The remaining 129 articles were included in the screening process. After reviewing the titles and abstracts, 102 articles were excluded due to inconsistencies with our study inclusion criteria. The full texts of the remaining 27 articles were further reviewed. Then, 20 articles were excluded based on the exclusion criteria. One study was excluded because it utilized Chinese female subjects aged 20–40 years as the reference population for DXA evaluation [25]. This reference demographic does not align with the WHO’s recommended reference standards. Finally, six studies were selected for this systematic review and meta-analysis [17,19,21,22,38,39]. Figure 1 illustrates the study selection process of this systematic literature review according to the PRISMA-DTA guidelines.

### 3.2. Quality Assessment

The QUADAS-2 tool was used to assess the quality of the six included studies. Figure 2 shows a summary and graph of bias and applicability concerns. Most studies had a low risk of bias and low concern about applicability. In one study [38], a high risk of bias was observed in ‘patient selection’ due to the exclusive inclusion of female participants, with no representation of male subjects, whereas in the other two studies [19,39], the risk of bias remained uncertain owing to the skewed gender distribution, with a predominance of female participants. There was a high risk of applicability concern observed in Zhang et al.’s study [25], as T-scores were calculated from BMD datasets of young Chinese women aged 20 to 40.

### 3.3. Characteristics of Included Studies

Six studies were conducted in Asia: three in Korea, two in Taiwan, and one in Japan. All these studies used a retrospective approach. Four were single-center studies and two were multicenter studies. Among these studies, the men-to-women ratio was imbalanced, with the majority of participants being female, and one study contained only female participants. The study included plain dental panoramic (N = 1), chest (N = 2), pelvic (N = 3), and lumbar (N = 1) radiographs. The total number of sets of image types was seven because Hsieh et al. investigated both pelvic and lumbar radiographs. In the included studies, deep learning methods were used as the index test and DXA-derived BMD served as the reference standard. All studies diagnosed osteoporosis using DXA as the reference standard, based on the WHO criteria (T-score ≤ −2.5), and compared the T-score with that of young white females as a reference value. Various deep learning models were employed in these studies: three utilized VGG16 [17,22,38], one employed ResNet18 [21], another utilized ResNet50 [39], and one adopted the OsPor-screen model [19]. Table 1 (at the end of the document) summarizes the characteristics of the included studies.

### 3.4. Descriptive Statistics and Diagnostic Accuracy

A meta-analysis was performed based on the results of a systematic review of six included studies. The research variables included various image types (dental, chest, pelvic, and lumbar radiographs), deep learning models (VGG16, ResNet18 and 50, and the OsPor-screen model) and participant populations (only female or both sexes). The study conducted by Hsieh et al. [22] achieved the highest performance, with an AUROC of 0.97, while Jang et al.’s [38] study had the lowest AUROC at 0.7.

A coupled forest plot of the specificity and sensitivity of deep learning models in this study with an appropriate 95% confidence interval (CI) is shown in Figure 3A. Sensitivity values for deep learning models in the present study ranged between 0.72 (95% CI: 0.67–0.76) and 0.90 (95% CI: 0.80–0.96), while specificity values ranged between 0.74 (95% CI: 0.60–0.77) and 0.95 (95% CI: 0.94–0.96). The highest sensitivity was conducted by Lee et al. using dental X-ray, and the highest specificity was conducted by Hsieh et al. using pelvic X-ray.

### 3.5. Threshold Effect and Heterogeneity

Figure 3B demonstrates the HSROC curve plot. A substantial deviation of individual results from the HSROC curve was observed, indicating moderate heterogeneity in the present study.

Descriptive analyses were conducted for subgroup analysis due to the limited number of studies included. However, as there was only one study each for dental and lumbar X-ray images, subgroup analyses for these categories were not feasible. Sensitivities for osteoporosis inference from pelvic and chest X-rays were 0.72–0.80 and 0.81–0.84, respectively, and the corresponding specificities were 0.86–0.95 and 0.74–0.81 (Figure 4). Sensitivity and meta-regression analyses were not performed because there were an insufficient number of studies available for analysis.

## 4. Discussion

Based on the meta-analysis of the six included studies, this research aimed to assess the diagnostic accuracy of deep learning methods in identifying osteoporosis from plain X-ray images. Our analysis revealed that deep learning achieved a pooled AUROC of 0.88 (95% CI 0.85–0.91), a sensitivity of 0.81 (95% CI 0.78–0.84), and a specificity of 0.87 (95% CI 0.81–0.92), indicating good diagnostic performance (0.8–0.9) [40]. These results indicated that deep learning methods can effectively extract bone density information from X-rays and the aforementioned performance suggests their utility for opportunistic screening.

A meta-analysis published in 2021 analyzed the performance of artificial intelligence-based systems in diagnosing osteoporosis from medical images, and the results showed a high degree of diagnostic accuracy, with an AUROC of 0.93, a sensitivity of 0.86 to 1.00, and a specificity of 0.75 to 1.00 [41]. However, this meta-analysis included studies with a variety of image types (X-ray and CT), AI models (machine learning and deep learning), diagnostic references (DXA, skeletal BMD examinations, and judgments made by oral and maxillofacial radiologists), and reference standards (T-score and eroded or normal mandibular cortex). Therefore, the lack of adequate analyses of deep learning methods for plain X-ray images makes it difficult to determine their potential for clinical implementation. To address this issue, the present study focused on the application of deep learning methods to plain X-ray images. Deep learning is efficient, automated, and scalable, and X-rays offer lower radiation, faster procedures, and cost-effectiveness. Together, these attributes can increase osteoporosis detection rates and at a low cost, making this approach essential for future clinical implementation. Although the number of included studies was limited, the results have already shown that plain radiography has good performance for inferring bone density.

We had planned to perform statistical subgroup analysis to compare performance across different experimental conditions. The present meta-analysis included two covariates: image type (dental, chest, pelvic, and lumbar radiographs) and participant population (only female or both sexes). However, the number of studies that used different types of radiographic images was too small to allow for statistical comparisons. Furthermore, several deep learning methods were applied in these studies, but these studies did not provide details of the deep learning neural networks, making it difficult to compare their performances. Future research should focus on expanding the dataset with more studies and standardizing methodologies, particularly in the image types and deep learning models used, to enhance the robustness and comparability of findings.

Although the performance of deep learning methods was quite promising in these studies, further development and optimization are required before successful clinical adoption. Future studies may consider replacing CNNs with transformer models, given their enhanced accuracy, superior performance in handling noisy or augmented images, and greater efficiency in computational resource usage and training time reduction [42]. Furthermore, transformers provide a complete understanding of entire images, in contrast to CNNs, which mainly focus on local feature relationships, thereby enabling more thorough information processing. Another intriguing approach could involve incorporating clinical covariates into our methodology. These covariates include factors such as age, sex, body mass index, and additional risk factors like previous fractures, current smoking, and femoral neck BMD, all of which are components of the FRAX tool [43]. Previous studies investigated whether the addition of clinical covariates enhances the diagnostic performance of image-only models. Yamamoto et al. discovered that incorporating clinical covariates like age, sex, and body mass index into a pelvic X-ray-based model enhanced its osteoporosis detection capabilities. This was evidenced by a 0.005 increase in the AUC, from 0.887 to 0.892 [23,24]. Despite not being included in our current meta-analysis due to the lack of provided sample size and confusion matrix details in their model, Yamamoto’s study offers a significant insight. Their methodology, which incorporates various relevant clinical data, has been shown to enhance the rate of osteoporosis detection. Furthermore, in clinical settings, the primary focus shifts from solely assessing BMD to a more critical aspect of predicting fracture likelihood. This shift is crucial as it aligns more closely with clinical decision-making regarding necessary interventions, emphasizing a more patient-centric approach in osteoporosis management. To this end, future deep learning studies should further predict fracture risk while incorporating relevant clinical variables [23].

An important issue is how deep learning methods can be incorporated into screening programs in real-world clinical settings. In a proposed opportunistic screening process, patients initially receive X-ray examinations for assorted reasons. If the deep learning analysis identifies a risk of osteoporosis, clinicians could then refer these patients for further DXA scans to confirm the diagnosis. This screening pipeline provides a unique opportunity for the early detection of osteoporosis as the frequency of undergoing plain radiography increases annually in older adults [44]. Using these deep learning methods, patients who are unaware of their osteoporosis can be recommended to undergo DXA, which can result in a significant increase in diagnostic rates. Therefore, early intervention and fracture prevention can be facilitated, thereby reducing fracture-associated burden and improving healthcare finance [45].

This study had a few limitations. First, the data of the confusion matrix could not be retrieved from several studies as these studies did not provide adequate information to yield the confusion matrix. Although we tried to contact the authors, only authors in some of these studies responded and provided the information; therefore, these studies were excluded and thus limited the sample size. Second, the included studies applied a variety of deep learning methods and image types, which may have introduced a high degree of methodological bias across studies, thereby affecting generalizability. Specifically, given that deep learning has been evolving at a fast pace in recent years, it is possible that studies using more advanced methods could yield a better result. Moreover, the heterogeneity observed in the HSROC would limit the reliability to yield a solid conclusion. Third, a few studies did not include external validation datasets and only offered results for their internal validation data, which could have led to an overfitting of the diagnostic accuracy of the algorithm. Finally, all included studies were conducted in Asia, which could limit their applicability to non-Asian populations. Indeed, the prevalence of osteoporosis in Asia is notably higher than in the USA and Australia [46], a geographic variability that highlights the need for a broader understanding of osteoporosis as well as the development of deep learning models that could fit different ethnic groups and different countries. To address these limitations in future research, it is recommended to ensure more comprehensive data reporting, utilize consistent and advanced deep learning methodologies, incorporate external validation datasets, and expand the geographic scope of studies to enhance applicability and generalizability.

## 5. Conclusions

This study demonstrates the potential of deep learning methods that use plain X-rays to detect osteoporosis. This approach improves the early detection of decreased bone mineral density, aiding clinicians in planning prompt interventions and reducing the risk of osteoporosis-related fractures and complications. However, owing to the limited number of available studies, further research is required to explore their utility in clinical applications.

## Figures and Tables

**Figure 1 diagnostics-14-00207-f001:**
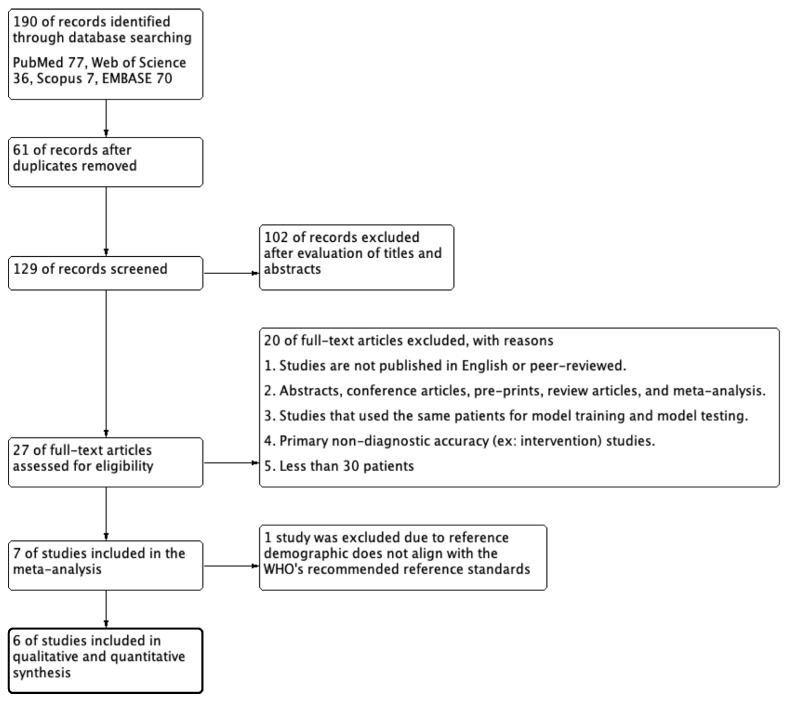
Flowchart of the systematic literature search according to PRISMA guidelines.

**Figure 2 diagnostics-14-00207-f002:**
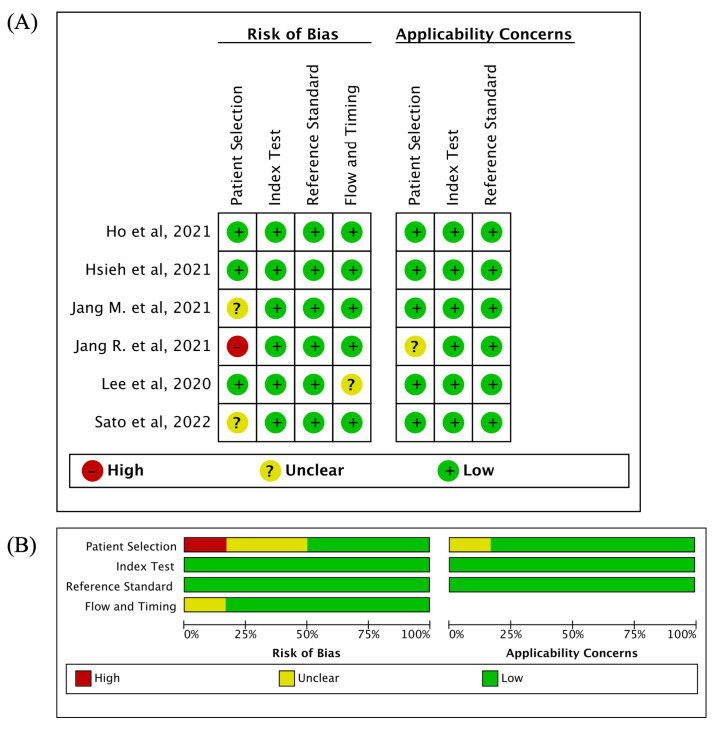
Summary and graph of bias and applicability concerns (**A**) Risk of bias summary (**B**) Risk of bias graph [17,19,21,22,38,39].

**Figure 3 diagnostics-14-00207-f003:**
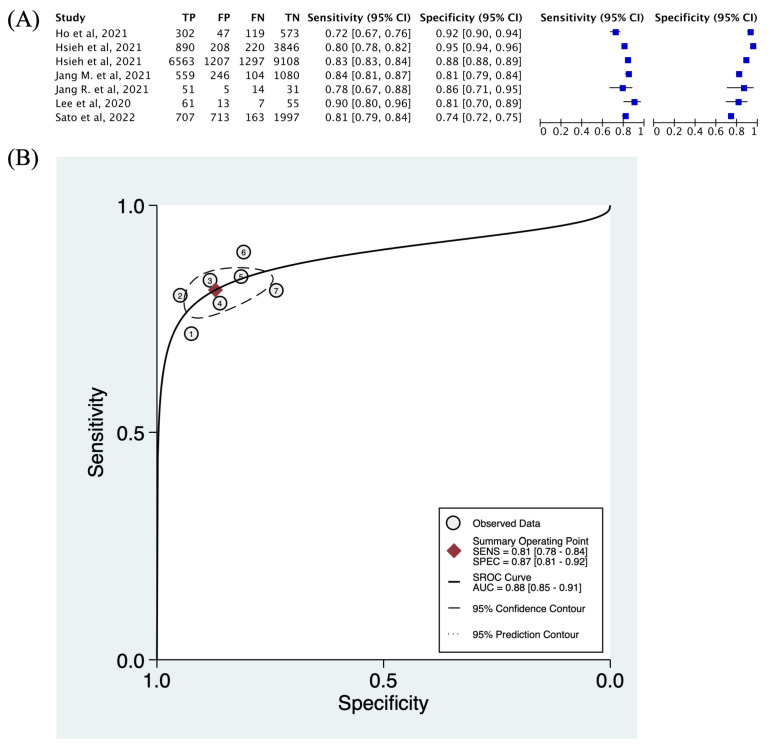
The forest plot and hierarchical summary receiver operating characteristic (HSROC) curve of included studies: (**A**) sensitivity and specificity; (**B**) HSROC curve of deep learning methods for detecting osteoporosis from X-ray images. Number 1–7 indicates: Ho et al., 2021 [21], Hsieh et al., 2021 (pelvic) [22], Hsieh et al., 2021 (lumbar) [22], Jang, R., 2021 [38], Jang, M., 2021 [19], Lee, 2020 [17], Sato et al., 2022 [39].

**Figure 4 diagnostics-14-00207-f004:**
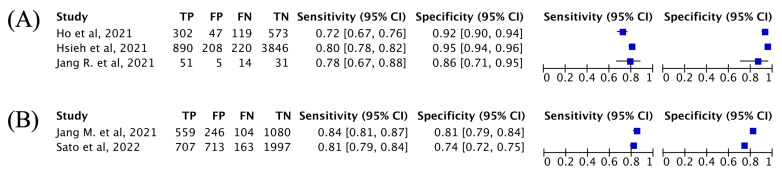
Forest plots of deep learning methods using (**A**) pelvic and (**B**) chest X-rays [19,21,22,38,39].

**Table 1 diagnostics-14-00207-t001:** Characteristics of included studies.

Author	Publication Year	Study Design/Center	Country	Medical Images	Reference Diagnosis	Reference Standard						Deep Learning Model
Ho et al. [21]	2021	Retrospective/Single-center	Taiwan	Pelvic X-ray	DXA	Osteoporosis and non-osteoporosis (cutoff T-score = −2.5 SD)						ResNet18
Hsieh et al. [22]	2021	Retrospective/Single-center	Taiwan	Pelvic X-ray	DXA	Osteoporosis and non-osteoporosis (cutoff T-score = −2.5 SD)						VGG16
	2021	Retrospective/Single-center	Taiwan	Lumbar X-ray	DXA	Osteoporosis and non-osteoporosis (cutoff T-score = −2.5 SD)						VGG16
Jang, R. et al. [38]	2021	Retrospective/Single-center	Korea	Pelvic X-ray	DXA	Osteoporosis (T-score ≤ −2.5) and osteopenia (T-score between −1.0 and −2.5)						VGG16
Jang, M. et al. [19]	2022	Retrospective/Single-center	Korea	Chest X-ray	DXA	Normal (T-score ≥ −1.0), osteopenia (−2.5 < T-score < −1.0), and osteoporosis (T-score ≤ −2.5)						OsPor-screen model
Lee et al. [17]	2020	Retrospective/Single-center	Korea	Dental X-ray	DXA	Osteoporosis and non-osteoporosis (cutoff T-score = −2.5 SD)						VGG16
Sato et al. [39]	2022	Retrospective/ Multi-center	Japan	Chest X-ray	DXA	Normal (T-score above −1.0), osteopenia (T-score between −1.0 and −2.5), and osteoporosis (T-score below −2.5)						ResNet50
**Study ID**		**Image Number**	**Training/Validation/Test/External Dataset**			**TP**	**FP**	**FN**	**TN**	**Sensitivity**	**Specificity**	**AUROC**
Ho et al. [21]		5027	3972/1041/0/0			302	47	119	573	0.717	0.924	NA
Hsieh et al. [22]		10,797 (hip)	5633/0/5164/2060			890	208	220	3846	0.802	0.949	0.97
		25,482 (spine)	7307/0/18175/3346			6563	1207	1297	9108	0.835	0.883	0.92
Jang, R. et al. [38]		1001	800/100/101/117			51	5	14	31	0.785	0.861	0.7
Jang, M. et al. [19]		14,115	9825/1212/1989/1089			559	246	104	1080	0.843	0.815	0.91
Lee et al. [17]		680	544 (split to 4:1 for training and validation)/136/0			61	13	7	55	0.897	0.809	0.858
Sato et al. [39]		17,899	12529/1790/3580/0			707	713	163	1997	0.813	0.737	0.84

Note: DXA, dual X-ray absorptiometry; DCNN, deep convolutional neural networks; VGG, visual geometry group; ResNet, residual neural network; TP, true positive; FP, false positive; FN, false negative; TN, true negative.

## Data Availability

All data generated or analyzed during this study are included in this published article.

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
