# Peer review of "Diagnostic Accuracy of Deep Learning for the Prediction of Osteoporosis Using Plain X-rays: A Systematic Review and Meta-Analysis"

_diagnostics, 2024, doi:10.3390/diagnostics14020207_

Round 1
Reviewer 1 Report
Comments and Suggestions for Authors
Lines 45 and 47: use "greater or equal" sign instead of ">".
Line 223: why are you using different names for the deep learning model adopted in [17] (here "Inception-V3" and in Table 1 "OsPor-screen model")?
Comments on the Quality of English Language
Line 41: insert space before "[2]".
Lines 238 and 239: "x-ray" should be "X-ray".
Line 241: "Hierarchical" should be "hierarchical".
Line 357 and later: use big letters in journal names. Example: Journal of Bone and Mineral Research.
Author Response
For review article
Response to Reviewer 1 Comments
1. Summary
Thank you very much for taking the time to review this manuscript. Please find the detailed responses below and the corresponding revisions/corrections highlighted/in track changes in the re-submitted files.
2. Point-by-point response to Comments and Suggestions for Authors
Comments 1: Lines 45 and 47: use "greater or equal" sign instead of ">".
Response 1: Thank you for pointing this out. We have change the ">" sign to the “≥” sign. (Line 43-46.)
According to the World Health Organization (WHO), osteoporosis is diagnosed by measuring bone mineral density (BMD) of the femoral neck using dual-energy X-ray absorption (DXA) [3] and is defined as BMD > 2.5 standard deviations (SDs) below the average value of the young white-female reference population (T-score ≤ -2.5), whereas low bone mass (or osteopenia) is defined as ≥ 1.0 SD but < 2.5 SDs below the average value of the young white-female reference population (-2.5 < T-score ≤ -1.0).
Comments 2: Why are you using different names for the deep learning model adopted in [17] (here "Inception-V3" and in Table 1 "OsPor-screen model")?
Response 2: Thank you for pointing this out. We have specifically changed "Inception-V3" to “OsPor-screen model”. (Line 226)
“[Various deep learning models were employed in these studies: three utilized VGG16 [16, 21, 37], one employed ResNet18 [20], another utilized ResNet50 [38], and one adopted OsPor-screen model [18].]”
3. Response to Comments on the Quality of English Language
Point 1: Line 41: insert space before "[2]".
Point 2: Lines 238 and 239: "x-ray" should be "X-ray".
Point 3: Line 241: "Hierarchical" should be "hierarchical".
Point 4: Line 357 and later: use big letters in journal names. Example: Journal of Bone and Mineral Research.
Response: We have made the necessary revisions based on the reviewer's suggestions as outlined above."
Response 1: Line 38
Fragility fractures increase morbidity and mortality in individuals and impose a huge financial burden on society as a whole [2].
Response 2: Line 238-240
Figure 3. The Forest plot and hierarchical summary receiver operating characteristic (HSROC) curve of included studies
Response 3: Line 234-236
The highest sensitivity conducted by Lee et al using dental X-ray, and the highest specificity was conducted by Hsieh et al using pelvic X-ray.
Response 4: Line 373 and later. We have revised as your suggestion.

Reviewer 2 Report
Comments and Suggestions for Authors
The idea of this meta-analysis is excellent. Still I do recommend the exclusion of Zhang study, since the reference for DXA evaluation were young adults aged 20 to 40 - so the diagnostic of osteoporosis was biased = lower diagnostic rates *since the correct reference range are young adults aged 20 to 30. Since it is 1/7 manuscripts, the findings will bias the entire cohort.
Author Response
For review article
Response to Reviewer 2 Comments
1. Summary
Thank you very much for taking the time to review this manuscript. Please find the detailed responses below and the corresponding revisions/corrections highlighted/in track changes in the re-submitted files.
3. Point-by-point response to Comments and Suggestions for Authors
Comments 1: The idea of this meta-analysis is excellent. Still I do recommend the exclusion of Zhang study, since the reference for DXA evaluation were young adults aged 20 to 40 - so the diagnostic of osteoporosis was biased = lower diagnostic rates *since the correct reference range are young adults aged 20 to 30. Since it is 1/7 manuscripts, the findings will bias the entire cohort.
Response 1: Thank you for bringing this importance issue up to use. We agree with this comment, and, therefore, we have excluded Zhang’s study in the revised manuscript. Generally speaking, the conclusions are the same after the exclusion of Zhang’s study.

Reviewer 3 Report
Comments and Suggestions for Authors
A very well executed paper
Please find my comment / suggestions
Introduction
It would be beneficial to clearly outline the specific research questions or hypotheses that the authors intend to address through the systematic review and meta-analysis.
The paper emphasizes the role of deep learning in medical imaging but lacks a brief discussion on why deep learning is particularly suitable for this task compared to traditional methods. Providing a concise rationale for choosing deep learning over other approaches could strengthen the introduction.
When citing previous studies, it would be helpful to include key statistical metrics such as sensitivity, specificity, positive predictive value, and negative predictive value, especially when discussing the diagnostic accuracy of various methods.
The introduction briefly mentions CT and MRI as methods for estimating BMD, but it would be beneficial to explicitly state how the proposed use of plain radiography with deep learning compares to these existing methods in terms of accuracy, cost-effectiveness, and practicality.
Methodology
There is some repetition of information in the section, particularly in the explanation of the search strategy and the inclusion and exclusion criteria. The information about the search strategy and criteria could be consolidated and presented more succinctly to avoid redundancy.
While the search terms are provided, it would be beneficial to include additional details such as the specific Boolean operators used, any truncation or wildcards, and any variations in the search strategy for different databases. This level of detail helps readers understand the rigor of the literature search.
The methodology mentions that disagreements regarding study inclusion were resolved through discussion, but it would be helpful to elaborate on the criteria or guidelines used for reaching a consensus. Providing transparency on the resolution process adds credibility to the study.
The methodology appropriately mentions that traditional methods like Cochran's Q test and Higgins' I2 statistics are not suitable for diagnostic test accuracy studies. However, it would be helpful to explain in more detail how the HSROC plot and correlation analysis were used to assess heterogeneity.
Discussion
The discussion starts by reporting the findings of the study, such as the pooled AUROC, sensitivity, and specificity of deep learning methods. However, it might be beneficial to explicitly mention that these results are based on the meta-analysis of the seven included studies, reinforcing the source of these findings.
The study appropriately compares its findings with a meta-analysis from 2021, highlighting the differences in methodology and focus. However, the discussion could delve deeper into the implications of these differences, explaining why the focus on plain X-ray images is crucial for future clinical implementation.
The discussion mentions the planned subgroup analysis but highlights the challenges due to a small number of studies with different image types and unspecified deep learning models. While acknowledging these challenges, consider providing insights into potential future directions or methodologies to address these limitations in future research.
The discussion recognizes the promising performance of deep learning methods but appropriately emphasizes the need for further development and optimization before clinical adoption. It might be valuable to briefly discuss potential avenues for improvement or optimization, such as refining neural network architectures or incorporating additional clinical data.
The paper discusses the potential clinical application of deep learning methods for opportunistic screening and early detection of osteoporosis. However, it could elaborate further on the practical implications and challenges of implementing such a screening pipeline in real-world clinical settings. Discussing the potential workflow and integration into existing healthcare systems would enhance the practicality of the proposed approach.
The discussion mentions previous studies that investigated the inclusion of clinical covariates to enhance diagnostic performance. Consider expanding on the implications of these findings and how future studies might benefit from incorporating relevant clinical variables for more comprehensive risk assessment.
The limitations are appropriately acknowledged, including challenges in obtaining data from some studies, methodological biases, potential overfitting, and the geographical limitation to Asian populations. Consider providing brief suggestions or considerations for addressing these limitations in future research.
The discussion touches upon the limitation of the study being conducted in Asia, but it could further discuss the generalizability of the findings to non-Asian populations. Consider providing insights into potential variations in osteoporosis prevalence or characteristics across different populations.
Comments on the Quality of English LanguageThere are complex and lengthy sentences through out the paper
It will be beneficial to perform an English roof reading
Author Response
For review article
Response to Reviewer 3 Comments
1. Summary
Thank you very much for taking the time to review this manuscript. Please find the detailed responses below and the corresponding revisions/corrections highlighted/in track changes in the re-submitted files.
2. Point-by-point response to Comments and Suggestions for Authors
Introduction
Comments 1: It would be beneficial to clearly outline the specific research questions or hypotheses that the authors intend to address through the systematic review and meta-analysis.
Response 1: Thank you for pointing this out. We have clearly defined our research objectives in the manuscript. (Line 99-102)
In the present meta-analysis, we aimed to (1) determine the diagnostic accuracy of plain radiography using deep learning models, and (2) assess the factors that determine the diagnostic accuracy for osteoporosis.
Comments 2: The paper emphasizes the role of deep learning in medical imaging but lacks a brief discussion on why deep learning is particularly suitable for this task compared to traditional methods. Providing a concise rationale for choosing deep learning over other approaches could strengthen the introduction.
Response 2: Thank you for pointing this out. We have added a brief discussion. Line 64-68
In particular, deep learning methods are increasingly preferred over traditional machine learning techniques, mainly due to their superior capability for efficient and automatic extraction of clinically relevant features. Moreover, their exceptional performance in processing and learning from vast, complex, and unstructured datasets further establishes deep learning as a more effective approach.
Comments 3: When citing previous studies, it would be helpful to include key statistical metrics such as sensitivity, specificity, positive predictive value, and negative predictive value, especially when discussing the diagnostic accuracy of various methods.
Response 3: Thank you for pointing this out. We have added key statistical metrics when citing previous studies. Line 80-93
Lee et al. found that a CNN-based computer-aided system analyzing panoramic X-rays was highly effective in detecting osteoporosis, outperforming oral and maxillofacial radiologists with an AUC of 0.858, sensitivity of 0.9, specificity of 0.815, and an accuracy of 0.84. [16]. Wang et al. developed a method that automatically inferred the BMD of the lumbar spine from chest radiographs and showed a good correlation (R = 0.84) between the predicted and ground-truth BMD values, with an AUROC of 0.936 for detecting osteoporosis [18]. Ho et al. reported that DeepDXA, a convolutional neural network regression model analyzing pelvic X-rays, demonstrates high accuracy in diagnosing osteoporosis, showing a strong correlation (R = 0.85) between its predictions and the actual BMD measurements [19]. Zhang et al. developed a deep learning model that effectively predicts osteoporosis and osteopenia in postmenopausal women by analyzing lumbar spine X-rays, achieving an AUC of 0.767 and a sensitivity of 73.7% [23]. These studies demonstrate that plain X-rays combined with deep learning can be used for the opportunistic screening of osteoporosis.
Comments 4: The introduction briefly mentions CT and MRI as methods for estimating BMD, but it would be beneficial to explicitly state how the proposed use of plain radiography with deep learning compares to these existing methods in terms of accuracy, cost-effectiveness, and practicality.
Response 4: Thank you for pointing this out to us. We have revised according to your suggestion. Line 72-77
Although research has revealed that using artificial intelligence on CT and MRI scans for BMD estimation is comparably effective to plain radiographs, with CT showing a correlation of 0.84 and an AUC of 0.97 in diagnosing osteoporosis [15], and MRI demonstrating a correlation of 0.64 [16], these methods are less practical for screening. This is because their high costs, approximately 10 to 15 times higher than plain X-rays in Taiwan, along with limited availability and longer both in awaiting and examination time.
Methodology
Comments 5: There is some repetition of information in the section, particularly in the explanation of the search strategy and the inclusion and exclusion criteria. The information about the search strategy and criteria could be consolidated and presented more succinctly to avoid redundancy.
Response 5: Agree. We have removed repetition of information and consolidated the search strategy. Line 125-132
Inclusion criteria for the studies were: (1) use of deep learning methods on X-ray images for osteoporosis detection or bone density estimation, (2) utilization of DXA as the reference standard method for diagnosing osteoporosis, and (3) containing information about the sample sizes of the test dataset. Exclusion criteria were: (1) non-English or not peer-reviewed studies; (2) abstracts, conference articles, preprints, review articles, and meta-analyses; (3) articles using the same patients for model training and model testing; (4) primary non-diagnostic accuracy (for example, intervention); and (5) less than 30 participants in each of the training and testing datasets.
Comments 6: While the search terms are provided, it would be beneficial to include additional details such as the specific Boolean operators used, any truncation or wildcards, and any variations in the search strategy for different databases. This level of detail helps readers understand the rigor of the literature search.
Response 6: Agree. Line 111-117
The search terms employed included "deep learning," "convolutional neural network," "CNN," "DCNN," "osteoporosis," "osteopenia," "bone mineral density," "BMD," "low BMD," along with various X-ray types such as "radiographs," "X-rays," "diagnostic X-rays," "chest X-rays," "pelvic X-rays," "lumbar spine X-rays," and "dental X-rays." Boolean operators like "AND" and "OR" were employed for greater search accuracy, and the scope was broadened through the use of truncations such as "radiograph*," "osteoporos*," and "diagnos*."
Comments 7: The methodology mentions that disagreements regarding study inclusion were resolved through discussion, but it would be helpful to elaborate on the criteria or guidelines used for reaching a consensus. Providing transparency on the resolution process adds credibility to the study.
Response 7: Agree. We resolve through discussion guided by the PRISMA guideline. Line 120-122
Disagreements regarding study inclusion were resolved through discussion guided by the PRISMA guideline, and a third investigator (Y.P.C) was consulted if a consensus could not be reached.
Comments 8: The methodology appropriately mentions that traditional methods like Cochran's Q test and Higgins' I2 statistics are not suitable for diagnostic test accuracy studies. However, it would be helpful to explain in more detail how the HSROC plot and correlation analysis were used to assess heterogeneity.
Response 8: Thank you for pointing this out. We have explained that evidence of heterogeneity was identified when individual studies showed significant deviation from the HSROC curve and the correlation coefficient between sensitivity and specificity exceeded zero. Line 175-177.
Heterogeneity was identified by visually observing the asymmetry of the SROC curve and the pronounced scattering of data points from individual studies along this curve [35, 37].
Discussion
Comments 9: The discussion starts by reporting the findings of the study, such as the pooled AUROC, sensitivity, and specificity of deep learning methods. However, it might be beneficial to explicitly mention that these results are based on the meta-analysis of the seven included studies, reinforcing the source of these findings.
Response 9: Following Reviewer 2's recommendation, we have excluded Zhang's study from our analysis. The reason for this exclusion is that the study's reference population for DXA evaluations primarily consisted of young adults aged 20 to 40. This age range could introduce a bias in the results. As a result of this exclusion, our analysis now encompasses only six studies. Line 255-257
Based on the meta-analysis of the six included studies, this research aimed to assess the diagnostic accuracy of deep learning methods in identifying osteoporosis from plain X-ray images.
Comments 9: The study appropriately compares its findings with a meta-analysis from 2021, highlighting the differences in methodology and focus. However, the discussion could delve deeper into the implications of these differences, explaining why the focus on plain X-ray images is crucial for future clinical implementation.
Response 9: Agree. Line 273-279
To address this issue, the present study focused on the application of deep learning methods to plain X-ray images. Deep learning is efficient, automated, and scalable, and X-rays offer lower radiation, faster procedures, and cost-effectiveness. Together, these attributes can increase osteoporosis detection rates at a low cost, making this approach essential for future clinical implementation. Although the number of included studies was limited, the results have already showed that plain radiography has a good performance for inferring bone density.
Comments 9: The discussion mentions the planned subgroup analysis but highlights the challenges due to a small number of studies with different image types and unspecified deep learning models. While acknowledging these challenges, consider providing insights into potential future directions or methodologies to address these limitations in future research.
Response 9: Agree. Line 287-289
Future research should focus on expanding the dataset with more studies and standardizing methodologies, particularly in image types and deep learning models used, to enhance the robustness and comparability of findings.
Comments 10: The discussion recognizes the promising performance of deep learning methods but appropriately emphasizes the need for further development and optimization before clinical adoption. It might be valuable to briefly discuss potential avenues for improvement or optimization, such as refining neural network architectures or incorporating additional clinical data.
Response 10: Agree. Line 291-301
Although the performance of deep learning methods was quite promising in these studies, further development and optimization are required before successful clinical adoption. To further enhance diagnostic accuracy, future studies may consider replacing CNNs with transformer models, given their enhanced accuracy, superior performance in handling noisy or augmented images, and greater efficiency in computational resource usage and training time reduction [42]. Furthermore, transformers provide a complete understanding of entire images, in contrast to CNNs, which mainly focus on local feature relationship, thereby enabling more thorough information processing. Another intriguing approach could involve incorporating clinical covariates into our methodology. These covariates include factors such as age, sex, body mass index, and additional risk factors like previous fractures, current smoking, and femoral neck BMD, all of which are components of the FRAX tool [43].
Comments 11: The paper discusses the potential clinical application of deep learning methods for opportunistic screening and early detection of osteoporosis. However, it could elaborate further on the practical implications and challenges of implementing such a screening pipeline in real-world clinical settings. Discussing the potential workflow and integration into existing healthcare systems would enhance the practicality of the proposed approach.
Response 11: Agree. Line 316-320
An important issue is how deep learning methods can be incorporated into screening programs in real-world clinical settings. In a proposed opportunistic screening process, patients initially receive X-ray examinations for assorted reasons. If the deep learning analysis identifies a risk of osteoporosis, clinicians could then refer these patients for further DXA scans to confirm the diagnosis.
Comments 12: The discussion mentions previous studies that investigated the inclusion of clinical covariates to enhance diagnostic performance. Consider expanding on the implications of these findings and how future studies might benefit from incorporating relevant clinical variables for more comprehensive risk assessment.
Response 12: Agree. Line 301-314
Previous studies investigated whether the addition of clinical covariates enhances the diagnostic performance of image-only models. Yamamoto et al. discovered that incorporating clinical covariates like age, sex, and body mass index into a pelvic X-ray-based model enhanced its osteoporosis detection capabilities. This was evidenced by a 0.005 increase in the AUC, from 0.887 to 0.892 [23, 24]. Despite not being included in our current meta-analysis due to the lack of provided sample size and confusion matrix details in their model, Yamamoto's study offers a significant insight. Their methodology, which incorporates various relevant clinical data, has shown to enhance the rate of osteoporosis detection. Furthermore, in clinical settings, the primary focus shifts from solely assessing BMD to a more critical aspect of predicting fracture likelihood. This shift is crucial as it aligns more closely with clinical decision-making regarding necessary interventions, emphasizing a more patient-centric approach in osteoporosis management. To this end, future deep learning studies should further predict fracture risk while incorporating relevant clinical variables [23].
Comments 13: The limitations are appropriately acknowledged, including challenges in obtaining data from some studies, methodological biases, potential overfitting, and the geographical limitation to Asian populations. Consider providing brief suggestions or considerations for addressing these limitations in future research.
Response 13: Agree. Line 344-348
To address these limitations in future research, it is recommended to ensure more comprehensive data reporting, utilize consistent and advanced deep learning methodologies, incorporate external validation datasets, and expand the geographic scope of studies to enhance applicability and generalizability.
Comments 14: The discussion touches upon the limitation of the study being conducted in Asia, but it could further discuss the generalizability of the findings to non-Asian populations. Consider providing insights into potential variations in osteoporosis prevalence or characteristics across different populations.
Response 14: Agree. Line 340-344
Finally, all included studies were conducted in Asia, which could limit their applicability to non-Asian populations. Indeed, the prevalence of osteoporosis in Asia is notably higher than in the USA and Australia [45], a geographic variability that highlights the need for a broader understanding of osteoporosis as well as the development deep learning models that could fit different ethnic groups and different countries.
4. Response to Comments on the Quality of English Language
Point 1: There are complex and lengthy sentences throughout the paper. It will be beneficial to perform an English roof reading
Response 1: We appreciate your feedback regarding the complexity and length of sentences in our paper. Following your recommendation, we have thoroughly reviewed and revised the manuscript to ensure clarity and conciseness in our language.

Round 2
Reviewer 3 Report
Comments and Suggestions for Authors
All my concerns have been addressed
congratulations to the authors
the paper can be accepted in current form